# AffinityFlow: Guided Flows for Antibody Affinity Maturation

**Can (Sam) Chen** [1 2]   **Karla-Luise Herpoldt** [2]   **Chenchao Zhao** [2]   **Zichen Wang** [2]   **Marcus Collins** [2]   **Shang Shang** [2]
**Ron Benson** [2]

## Abstract

Antibodies are widely used as therapeutics, but their development requires costly affinity maturation, involving iterative mutations to enhance binding affinity. This paper explores a sequence-only scenario for affinity maturation, using solely antibody and antigen sequences. Recently AlphaFlow wraps AlphaFold within flow matching to generate diverse protein structures, enabling a sequence-conditioned generative model of structure. Building on this, we propose an *alternating optimization* framework that **(1)** fixes the sequence to guide structure generation toward high binding affinity using a structure-based affinity predictor, then **(2)** applies inverse folding to create sequence mutations, refined by a sequence-based affinity predictor for post selection. A key challenge is the lack of labeled data for training both predictors. To address this, we develop a *co-teaching* module that incorporates valuable information from noisy biophysical energies into predictor refinement. The sequence-based predictor selects consensus samples to teach the structure-based predictor, and vice versa. Our method, *AffinityFlow*, achieves state-of-the-art performance in proof-of-concept affinity maturation experiments.

## 1. Introduction

Natural antibodies protect organisms by specifically binding to target antigens such as viruses and bacteria with high affinity (Murphy & Weaver, 2016), while therapeutic antibodies bind various targets to inactivate them, recruit immune cells to them, or deliver an attached drug compound (Chiu & Gilliland, 2016). *In vivo*, antibodies undergo affinity maturation, where their target-binding capacity is incrementally

---

[1]Mila - Quebec AI Institute (work done during an Amazon internship) [2]Amazon. Correspondence to: Can (Sam) Chen <can.chen@mila.quebec or chencan421@gmail.com>, Marcus Collins <collmr@amazon.com>.

*Proceedings of the 42^nd International Conference on Machine Learning*, Vancouver, Canada. PMLR 267, 2025. Copyright 2025 by the author(s).

enhanced through somatic hypermutation and clonal selection (Victora & Nussenzweig, 2022). When developing a therapeutic antibody, *in vitro* affinity maturation—through targeted mutation and selection—improves the binding affinity of existing antibodies to target antigens (Tabasinezhad et al., 2019; Chiu & Gilliland, 2016).

These *in vitro* methods, such as random mutagenesis, are labor-intensive and time-consuming. Recent advancements in deep learning have propelled *in silico* affinity maturation forward. One line of research enhances affinity prediction for mutated antibodies (Shan et al., 2022; Liu et al., 2021; Cai et al., 2024; Lin et al., 2024; Xiong et al., 2017); another investigates mutation strategies. Specifically, protein language models propose plausible mutations to enhance binding affinities, though they lack specificity for target antigens (Hie et al., 2024; Ruffolo et al., 2021; Shuai et al., 2021). Similarly, diffusion models guide the sampling towards high-affinity antibody sequences but require the often unavailable or insufficiently accurate antigen-antibody complex structure (Luo et al., 2022; Zhou et al., 2024). Our research aligns more with the second line of mutation strategies. In particular, we focus on enhancing antibody binding affinity through sequence mutations, relying solely on the antigen-antibody sequence.

Recognizing the crucial link between antibody structure and function, it is essential to integrate structure into the sequence mutation process. The recent release of AlphaFlow (Jing et al., 2024) builds a sequence-conditioned generative model of protein structure, which opens pathways for structure-based optimization of antibody sequences. Specifically, AlphaFlow repurposes AlphaFold (Jumper et al., 2021) in a flow matching framework to generate diverse protein conformations.

This motivates the proposal of an *alternating optimization* framework, as illustrated in Figure 1: **(1)** We fix the sequence to guide noisy structures toward high-affinity clean structures. Rather than re-training the entire AlphaFlow model—a process that is inherently time-consuming—we achieve guided structure generation through predictor guidance (Dhariwal & Nichol, 2021). Specifically, a trained structure-based affinity predictor is integrated into the AlphaFlow sampling process to direct coordinate denoising.

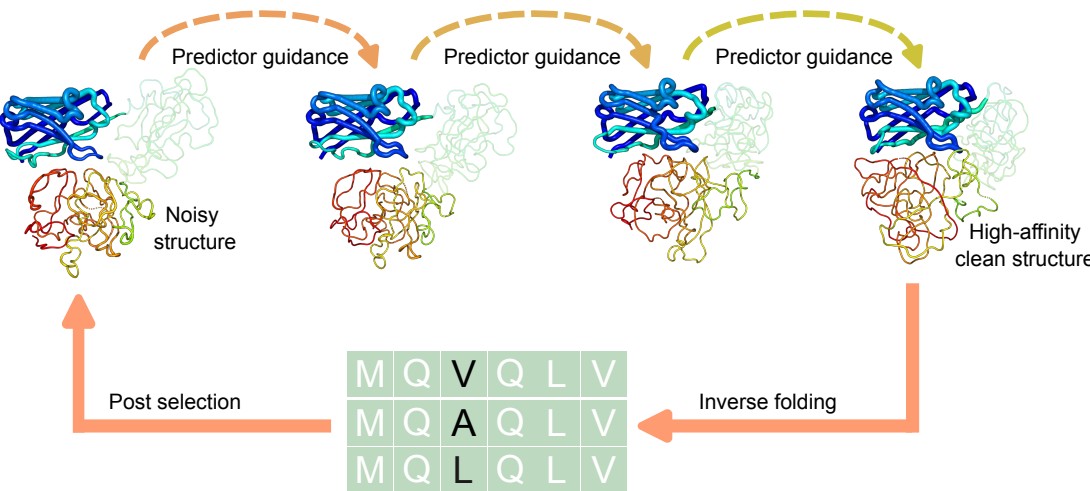

Figure 1: Illustration of *alternating optimization*.

**(2)** With the high-affinity clean structure, we perform inverse folding to introduce targeted mutations, and use a sequence-based predictor for post selection, which identifies promising mutated sequences for the next iteration.

A significant challenge in training both predictors is the scarcity of labeled data. To address this, we develop a *co-teaching* module that leverages valuable information from noisy biophysical energies to refine the predictors, as shown in Figure 2. For any antigen $i$ and antibodies $j, k, m, n$, we use Rosetta (Alford et al., 2017) to compute the binding free energy $\Delta G$ and then calculate the change in binding free energy $\Delta\Delta G_{ijk} = \Delta G_{ij} - \Delta G_{ik}$ to form pairwise discrete labels. The sequence-based predictor selects pairs with which it concurs, considering them likely to be accurate and informative, and uses these consensus samples to enhance the structure-based predictor. For instance, if the sequence-based predictor predicts $\Delta\Delta\hat{G}_{ijk} > 0$, it selects $\Delta\Delta G_{ijk} > 0$ for training the structure-based predictor. Similarly, the structure-based predictor reciprocates by informing the sequence-based predictor; for example, it selects $\Delta\Delta G_{ijm} < 0$ to refine the sequence predictor, as shown in Figure 2. Noisy data, such as $\Delta\Delta G_{ijn} > 0$, are filtered out. This module effectively integrates biophysical insights into both predictors, enhancing their accuracy.

In summary, we introduce *AffinityFlow*, guided flows for affinity maturation. Our contributions are three-fold:

- We present an AlphaFlow-based *alternating optimization* framework that guides structure generation towards high binding affinity through predictor guidance, followed by targeted mutations.

- We propose a *co-teaching* module that utilizes valuable insights from noisy biophysical energies to refine structure- and sequence-based predictors.

- *AffinityFlow* achieves state-of-the-art performance in affinity maturation experiments.

## 2. Preliminaries

### 2.1. Problem Definition

Binding affinity between an antibody (Ab) and its antigen (Ag) is predominantly determined by the complementarity determining regions (CDRs) within these chains (Akbar et al., 2022). An antibody consists of two heavy chains and two light chains with a similar overall structure. Its specificity is determined by six variable regions known as Complementarity Determining Regions (CDRs), denoted as H1, H2, H3, L1, L2, and L3. Typically, heavy chain CDRs range from 8 to 16 amino acids, while light chain CDRs range from 3 to 10 amino acids. We model an antibody chain as a sequence of amino acids, each characterized by a type $a_i \in \{A, C, D, \ldots, Y\}$.

While *AffinityFlow* is applicable to all antibody types, this study specifically focuses on single-domain antibodies (sdAb), which consist only of heavy chains (Wesolowski et al., 2009). We select sdAb for their high specificity, solubility, stability, and lower toxicity and immunogenicity. Our goal in affinity maturation is to effectively mutate the CDRs within the context of the entire Ab-Ag sequence complex to improve binding affinity.

### 2.2. Flow Matching

Flow matching is a robust generative modeling framework (Lipman et al., 2022; Le et al., 2024). It is characterized by a conditional probability path $p_t(\boldsymbol{x} \mid \boldsymbol{x}_1), t \in [0, 1]$, which transitions from a prior distribution $p_0(\boldsymbol{x} \mid \boldsymbol{x}_1) = q(\boldsymbol{x})$ to an approximate Dirac delta function $p_1(\boldsymbol{x} \mid \boldsymbol{x}_1) \approx \delta(\boldsymbol{x} - \boldsymbol{x}_1)$ conditioned on a data point $\boldsymbol{x}_1$ from $p_{\text{data}}$. The evolution

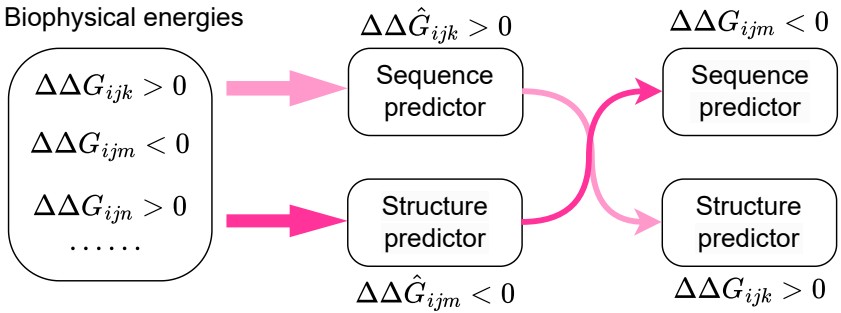

Figure 2: Illustration of *co-teaching*.

is facilitated by a conditional vector field $u_t(\boldsymbol{x} \mid \boldsymbol{x_1})$. The marginal vector field $v(\boldsymbol{x}, t)$ is modeled through a neural network parameterized by $\boldsymbol{\theta}$:

$$\hat{v}(\boldsymbol{x}, t; \theta) \approx v(\boldsymbol{x}, t) = \mathbb{E}_{\boldsymbol{x_1} \sim p_t(\boldsymbol{x_1}|\boldsymbol{x})}[u_t(\boldsymbol{x} \mid \boldsymbol{x_1})] \quad (1)$$

Using this modeled vector field, we can generate samples from the data distribution $p_{\text{data}}(\boldsymbol{x})$ by utilizing the corresponding neural Ordinary Differential Equation (ODE).

### 2.3. AlphaFlow

AlphaFold (Jumper et al., 2021) serves as a precise single-state protein structure predictor, and AlphaFlow (Jing et al., 2024) repurposes AlphaFold within a flow matching framework to generate diverse protein conformations. Given a protein sequence $\boldsymbol{a}$ of length $N$, the objective is to model the structural ensemble, denoted by $p(\boldsymbol{x} \mid \boldsymbol{a})$, where $\boldsymbol{x} \in \mathbb{R}^{3 \times N}$ represents the protein 3D coordinates.

AlphaFlow defines the conditional probability path by sampling initial noise $\boldsymbol{x_0}$ from $q(\boldsymbol{x_0})$ and linearly interpolating it with the data point $\boldsymbol{x_1}$:

$$\boldsymbol{x} \mid \boldsymbol{x_1}, t = (1 - t) \cdot \boldsymbol{x_0} + t \cdot \boldsymbol{x_1}, \quad \boldsymbol{x_0} \sim q(\boldsymbol{x_0}) \quad (2)$$

The vector field is derived as:

$$u_t(\boldsymbol{x} \mid \boldsymbol{x_1}) = (\boldsymbol{x_1} - \boldsymbol{x})/(1 - t) \quad (3)$$

Instead of directly parameterizing the marginal vector field as in Eq. (1), the marginal vector field is parameterized in terms of a neural network $\hat{\boldsymbol{x}}_1(\boldsymbol{x}, t; \boldsymbol{\theta})$ as:

$$\hat{v}(\boldsymbol{x}, t; \boldsymbol{\theta}) = (\hat{\boldsymbol{x}}_1(\boldsymbol{x}, t; \theta) - \boldsymbol{x})/(1 - t) \quad (4)$$

This approach allows the reuse of the AlphaFold2 template embedding stack to reconstruct the clean structure $\boldsymbol{x_1}$ from the noisy input $\boldsymbol{x}$, with $t$ serving as an additional time embedding. The model focuses on the 3D coordinates of $\beta$-carbons (or $\alpha$-carbon for glycine), defining the

prior distribution $q(\boldsymbol{x})$ over these positions as a harmonic prior (Jing et al., 2023) to ensure that inputs to the neural network remain physically plausible. For this study, the pretrained AlphaFlow model, which was trained using FAPE loss, is used directly without any further fine-tuning. Since AlphaFlow is trained solely on single proteins, this study connects the antibody sequence and the antigen sequence into one sequence using a linker of 32 *GGGGS* repeats (Lin et al., 2023). The linked sequence complex is then input into the system.

### 2.4. Affinity Prediction

This paper focuses on enhancing binding affinity, determined by the difference in free energy between the bound and unbound states, denoted $\Delta G$. A negative value indicates that the overall free energy of the system decreases upon binding, meaning that the antibody–antigen interaction is energetically favored. Consequently, we use $\Delta G$ as a measure of binding affinity where sequence- and structure-based predictors directly output a negative value of the binding affinity (Kd). Protein properties can be predicted from two views: sequence and structure, leading to two prediction methods: sequence-based (Xu et al., 2022) and structure-based (Gligorijević et al., 2021).

Leading sequence-based models like ESM-2 (Lin et al., 2022), AntiBERTy (Ruffolo et al., 2021), and IgLM (Shuai et al., 2023) are pre-trained on extensive unlabeled protein sequences. These models extract hidden representations to predict properties such as binding energy, denoted as $f_{\boldsymbol{\alpha}}(\Delta G|\boldsymbol{a})$, where $\boldsymbol{\alpha}$ represents the model parameters. We choose ESM-2 as our sequence-based predictor due to its versatility, as the antigen is a general protein rather than an antibody. Specifically, we input the antibody and antigen sequences separately into ESM-2 to obtain embeddings, which are then concatenated and fed into a three-layer MLP for the final prediction.

For structure-based prediction, the GVP model is notable for utilizing features from the 3D graph of proteins to predict properties, denoted as $f_{\boldsymbol{\beta}}(\Delta G | \boldsymbol{x})$ (Jing et al., 2021). Integrating the ESM2 model as a feature extractor within the GVP model further enhances performance (Wang et al., 2022). Thus, we employ the ESM2-GVP model as our structure-based predictor in this study. The linked antibody-antigen complex is processed by the pre-trained ESM2 to generate residue embeddings, from which intersected residues are selected for the GVP model.

It is important to note two aspects: (1) The ESM2-GVP model may not outperform the standalone ESM-2 model due to potential unavailability of ground-truth structures and the challenges in making reliable structure predictions for antibodies and antigens; (2) Given that our AlphaFlow system operates solely on $C_{\beta}$ coordinates and does not account for side-chains, we do not utilize affinity prediction models that require side chain modeling (Liu et al., 2021; Cai et al., 2024).

Related works on generative protein modeling and co-teaching are detailed in Appendix A.

## 3. Method

In this section, we introduce *AffinityFlow*, designed to enhance the binding affinity of antibodies through targeted sequence mutations. Our method, built on AlphaFlow, employs an *alternating optimization* framework for sequence mutation via predictor guidance, as detailed in Section 3.1. To overcome the challenge of limited labeled data, we propose a *co-teaching* module in Section 3.2. This module leverages useful knowledge from noisy biophysical energies to improve our predictors.

### 3.1. Alternating Optimization

Initially, the sequence is fixed while we guide the Ab structure generation to achieve high binding affinity, supplemented with predictor-corrector refinement. Based on the generated structure, we then use inverse folding to introduce targeted mutations into the Ab, with the sequence-based predictor selecting promising sequences for the next iteration.

**Guided Structure Generation**    While AlphaFlow generates structures unconditionally, we aim to steer structure generation toward improved binding affinity using predictor guidance. Following *Lemma 1* in (Zheng et al., 2023), predictor guidance in flow matching is formulated as:

$$\tilde{v}(\boldsymbol{x}_t, t, \Delta G; \boldsymbol{\theta}) = \hat{v}(\boldsymbol{x}_t, t; \boldsymbol{\theta}) + \frac{1-t}{t}\nabla_{\boldsymbol{x}_t}\log p_{\boldsymbol{\beta}}(\Delta G \mid \boldsymbol{x}_t, t). \tag{5}$$

where $p_{\boldsymbol{\beta}}(\Delta G \mid \boldsymbol{x}_t, t)$ denotes the target binding energy distribution. The derivation details are in Appendix B.

Training the predictor at different time steps $t$ is resource-intensive; instead, we approximate $p_{\boldsymbol{\beta}}(\Delta G \mid \boldsymbol{x}_t, t)$ directly from $p_{\boldsymbol{\beta}}(\Delta G \mid \hat{\boldsymbol{x}}_1(\boldsymbol{x}_t), 1)$:

$$p_{\boldsymbol{\beta}}(\Delta G \mid \boldsymbol{x}_t, t) \approx p_{\boldsymbol{\beta}}(\Delta G \mid \hat{\boldsymbol{x}}_1(\boldsymbol{x}_t), 1). \tag{6}$$

This approximation, termed $p_{\boldsymbol{\beta}}(\Delta G \mid \hat{\boldsymbol{x}}_1(\boldsymbol{x}_t))$, is effective when $t$ is close to 1; therefore, we primarily apply predictor guidance in the later stages of sampling.

The desired binding energy distribution is formulated as (Lee et al., 2023):

$$p_{\boldsymbol{\beta}}(\Delta G \mid \hat{\boldsymbol{x}}_1(\boldsymbol{x}_t)) = e^{-\gamma \hat{f}_{\boldsymbol{\beta}}(\hat{\boldsymbol{x}}_1(\boldsymbol{x}_t))}/Z, \tag{7}$$

where $\gamma$ is a scaling factor and $Z$ a normalization constant, with the negative sign indicating a preference for lower binding energy. Integrating this into Eq.(5) leads to:

$$\tilde{v}(\boldsymbol{x}_t, t, \Delta G; \boldsymbol{\theta}) = \hat{v}(\boldsymbol{x}_t, t; \boldsymbol{\theta}) - \gamma\frac{1-t}{t}\nabla_{\boldsymbol{x}_t}\hat{f}_{\boldsymbol{\beta}}(\hat{\boldsymbol{x}}_1(\boldsymbol{x}_t)). \tag{8}$$

This vector field guides the ODE sampling process towards lower binding energy. During sampling, we target the predictor guidance only to CDR coordinates rather than the full protein to simplify the system and enhance its relevance.

**Predictor-Corrector**    Given that $\hat{\boldsymbol{x}}_1$ represents the $C_{\beta}$ coordinates of the protein structure, which are subject to energy constraints, we apply Amber relaxation (Lindorff-Larsen et al., 2010) to adjust $\hat{\boldsymbol{x}}_1$ at each iteration before initiating guided structure generation. This correction step is essential, as predictor guidance on clashed protein structures is ineffective. This approach aligns with the Predictor-Corrector methods described in (Allgower & Georg, 2012; Song et al., 2020), and we therefore adopt the same terminology. Predictor corresponds to the protein coordinate generation process governed by the learned vector field, and Corrector refers to the Amber energy minimization used to refine the coordinates. Additional related techniques are discussed in Appendix C.

**Sequence Mutation**    Using the generated structure as a reference, we employ inverse folding with ProteinMPNN (Dauparas et al., 2022) to identify potential mutations in the CDR regions. At each iteration, we apply single-, double-, and triple-point mutations using ProteinMPNN. For each position, we calculate the probability difference between the current and alternative amino acids, selecting the mutation with the highest difference. Double- and triple-point mutations build sequentially on prior mutations. A sequence-based predictor selects the top $K$ ($K = 3$) sequences at each stage for further refinement. Since the generated structure is conditioned on the sequence, we avoid multiple simultaneous mutations to preserve the protein structure and minimize disruptive changes. However, under our *alternating optimization* framework, we can introduce a few mutations per

iteration, gradually accumulating enough mutations over successive iterations.

## 3.2. Co-teaching

A primary challenge is the scarcity of labeled data for training both structure-based and sequence-based affinity predictors. To address this, we enhance the predictors by incorporating insights from noisy biophysical energies.

**Complex Generation**    To compute biophysical energies, initial protein complexes are required. We extract $A$ sdAb structures and $B$ antigen structures from existing PDB files, and then use the docking tool GeoDock (Chu et al., 2023) to generate $AB$ complex structures. Next, we employ Rosetta (Alford et al., 2017) to calculate the binding free energy $\Delta G$ for each complex.

**Pairwise Discrete Data**    Instead of relying on pointwise continuous samples, which can be highly variable and noisy, we generate robust pairwise discrete data. For the $i$-th antigen, we pair antibody $j$ with antibody $k$ and compute the change in binding free energy as $\Delta\Delta G_{ijk} = \Delta G_{ij} - \Delta G_{ik}$. We assign a pairwise label $Y_{ijk}$ as 1 if $\Delta\Delta G_{ijk} > 0$, indicating stronger binding by antibody $k$, and 0 otherwise. This approach provides a more reliable measure than using absolute property values.

**Sample Selection**    Given the potential noise from unreliable biophysical energy calculations, we implement a reciprocal filtering approach to refine the quality of input for each predictor. Each predictor selects samples that align with its predictions to inform the other. Specifically, the sequence-based predictor $f_{\boldsymbol{\alpha}}(\Delta G|\boldsymbol{a})$ computes $\hat{Y}_{ijk}^a = (\Delta\Delta\hat{G}_{ijk} > 0)$. If $\hat{Y}_{ijk}^a = Y_{ijk}$, this indicates probable accuracy, prompting us to use this consensus sample for the structure-based predictor. The structure-based predictor undergoes a similar process, creating a cyclical filtering system. This ensures both predictors receive well-vetted, high-quality samples for improved reliability.

**Fine-tuning**    With the selected samples, we aim to enhance the performance of our predictors. For the sequence-based predictor, we minimize the following loss function:

$$
\begin{aligned}
\mathcal{L}(\boldsymbol{\alpha}) = -\sum_{i,j,k} \Big[ & Y_{ijk} \log p_{\boldsymbol{\alpha}}(Y_{ijk} = 1) \\
& + (1 - Y_{ijk}) \log(1 - p_{\boldsymbol{\alpha}}(Y_{ijk} = 1)) \Big],
\end{aligned} \tag{9}
$$

where $p_{\boldsymbol{\alpha}}(Y_{ijk} = 1) = \sigma(\Delta\hat{G}_{ij}^a - \Delta\hat{G}_{ik}^a)$ and $\sigma(\cdot)$ is the sigmoid function. The structure-based predictor undergoes a similar fine-tuning process. Through this *co-teaching* module, both predictors exchange valuable biophysical information, significantly improving their effectiveness.

## 4. Experiments

### 4.1. Benchmark

**Dataset**    We conduct our experiments using a sdAb subset of the SAbDab dataset (Dunbar et al., 2014). Following the protocol of (Luo et al., 2022), we exclude structures with a resolution poorer than 4Å and antibodies targeting non-protein antigens. Our study focuses on sdAbs, selecting PDB files of 120 labeled sdAb-antigen pairs to initially train our predictors using mean squared loss. From these files, we extract 77 sdAbs and 54 antigens, resulting in 4,158 docked complex structures generated by GeoDock. Rosetta is then used to calculate the $\Delta G$ for these complexes. For maturation testing, we select 60 sdAb-antigen PDB files, ensuring that each antigen is unique and these antigens and antibodies were not included in the training set.

**Evaluation**    Our evaluation considers mutations in CDR-H1, CDR-H2, CDR-H3, and the entire CDR region. Each comparative method generates three mutated sequences per antigen, resulting in a total of 180 sequence designs.

We measure performance using three metrics: functionality, specificity, and rationality, following (Ye et al., 2024). *Functionality* is assessed by the Improvement Percentage (*IMP*) as described in (Luo et al., 2022). IMP reflects the proportion of mutated sdAbs with reduced binding energy compared to the original. Structures are predicted using IgFold (Ruffolo et al., 2023), docked with GeoDock (Chu et al., 2023), and binding energies are analyzed via Rosetta (Alford et al., 2017). We report *IMP* instead of absolute values to ensure robustness, where a higher *IMP* indicates better performance. *Specificity* measures the sequence similarity among antibodies targeting different antigens. An effective method should generate distinct antibodies for different antigens, so lower similarity (*Sim*) indicates better specificity. *Rationality* is evaluated using inverse perplexity calculated by AntiBERTy (Ruffolo et al., 2021). This metric, also referred to as naturalness (*Nat*), indicates that higher values of *Nat* generally reflect better rationality.

### 4.2. Comparisons with Other Methods

In this paper, we primarily benchmark our method against language model-based methods, given our focus on sequence design. Since our method incorporates additional biophysical energies for training, we ensure fair comparisons by applying the same trained sequence-based predictor across all methods, unless stated otherwise. Each method generates a pool of candidate designs, and the sequence-based predictor selects the top three for final evaluation.

We consider the following language model-based methods:

1. **ESM (Hie et al., 2024)**: This method uses a pre-trained language model to identify potential mutations. Mutation consensus among six ESM models is assessed, and all promising sequences are collected over nine rounds.

2. **AbLang (Tobias H. Olsen & Deane, 2022)**: Specifically trained on antibody sequences, the AbLang model includes separate models for heavy and light chains. For our purposes, we utilize the heavy chain model to identify promising mutations across nine rounds.

3. **nanoBERT (Hadsund et al., 2024)**: Given our focus on sdAbs, nanoBERT, a model pre-trained on sdAb sequences, is employed. We conduct nine rounds of mutation identification.

Beyond language model-based methods, we include an additional sequence-design baseline:

4. **dWJS (Frey et al., 2023)**: handles discrete sequences by learning a smoothed energy function, sampling from the smoothed data manifold, and projecting the sample back to the true data manifold with one-step denoising.

We also evaluate three structure-based methods. Although our approach is sequence-based and does not inherently require structures for design, we use AlphaFold2 (Jumper et al., 2021) to predict the structures needed for these comparisons. The following methods are considered:

5. **DiffAb (Luo et al., 2022)**: Trains a diffusion model on amino acid types, coordinates, and orientations. Antibody optimization is achieved by introducing small perturbations into the existing antibody-antigen complex and subsequently denoising the structure. We generate ten designs per antigen and use our predictor to select the top three for evaluation.

6. **AbDPO (Zhou et al., 2024)**: Based on DiffAb, this model fine-tunes a pre-trained diffusion model using a residue-level decomposed energy preference to enable a low-energy protein sampling process. The sampling and selection processes are similar to those of DiffAb.

7. **GearBind (Cai et al., 2024)**: Utilizes multi-level geometric message passing and contrastive pretraining to improve predictions of affinity. We employ AbDPO to produce ten designs per antigen, from which GearBind selects the three most promising for assessment.

### 4.3. Training Details

We use a linker composed of 32 *GGGGS* repeats to connect the sdAb and antigen. Our method utilizes the *alternating optimization* framework with three iterations, where each

iteration introduces single-point, double-point, and triple-point mutations. This allows for producing 1 to 9 mutations in total. We set the AlphaFlow sampling steps $T$ to 3 per iteration with a schedule of $[1.0, 0.6, 0.3, 0.0]$ and use a default scaling factor $\gamma$ of 5. We employ ESM2-8M, followed by a hidden-dim-320 three-layer MLP, as the sequence-based predictor parameterized by $\boldsymbol{\alpha}$. For the structure-based predictor parameterized by $\boldsymbol{\beta}$, we use a five-layer GVP model, which takes the structure and ESM2-8M residue embeddings as input. For the co-teaching module, we use a batch size of 256 and a learning rate of $1 \times 10^{-4}$ with the Adam optimizer (Kingma, 2014). Computational efficiency is detailed in Appendix D, and hyperparameter sensitivity is addressed in Appendix E.

### 4.4. Results and Analysis

In Table 1, we present the experimental results on four settings CDR-H1, CDR-H2, CDR-H3 and all design positions. Delineating lines are drawn to distinguish between different groups of methods. The best and second-best performance are highlighted in **bold** and underlined, respectively.

We make the following observations: **(1)** As shown in Table 1, our method consistently achieves the best performance in terms of *IMP* and *Sim*, thereby highlighting its effectiveness. **(2)** The notable *IMP* is mainly due to our effective predictor guidance, which directs the structure sample generation towards low binding energy. **(3)** The low *Sim* scores can be attributed to antigen-specific modeling and the diversity introduced by the AlphaFlow sampling process. Language-based methods like ESM, AbLang, and nanoBERT lack this feature, as they do not incorporate specific antigens into their design processes. Structure-based methods such as DiffAb, AbDPO, and GearBind consider specific antigens, but their simplistic diffusion models are less effective at capturing antigen information compared to our method.

**(4)** The language model-based methods ESM, AbLang, and nanoBERT achieve the highest *Nat* scores, as they are implicitly trained for this metric. Beyond these methods, our approach achieves the best *Nat*. We attribute this to the realistic structure modeling enabled by AlphaFlow and the reliable inverse folding performed by ProteinMPNN, which together translate structures into natural sequences. **(5)** AbDPO, as a robust baseline, often achieves strong performance in *IMP*, likely due to incorporating energy information into its training, allowing for low-energy protein sampling. However, AbDPO requires training a separate diffusion model for each complex, adding complexity. **(6)** Lastly, the high *IMP* scores for baseline methods can largely be attributed to our trained sequence-based predictor. When using a standard predictor trained only on supervised data, *IMP* scores drop significantly. For example, in the CDR-H3

Table 1: Overall performance comparison

| Method | CDR-H1 | | | CDR-H2 | | | CDR-H3 | | | All | | |
|---|---|---|---|---|---|---|---|---|---|---|---|---|
| | IMP↑ | Sim↓ | Nat↑ | IMP↑ | Sim↓ | Nat↑ | IMP↑ | Sim↓ | Nat↑ | IMP↑ | Sim↓ | Nat↑ |
| ESM | 85.5% | 0.559 | **0.347** | 72.9% | 0.566 | **0.358** | 64.0% | 0.573 | **0.359** | 84.1% | 0.562 | **0.361** |
| AbLang | 88.0% | 0.536 | 0.330 | 85.4% | 0.537 | 0.322 | 88.5% | 0.542 | 0.336 | 82.9% | 0.548 | 0.349 |
| nanoBERT | 84.7% | 0.534 | 0.322 | 85.9% | 0.536 | 0.321 | 81.6% | 0.537 | 0.328 | 86.0% | 0.544 | 0.341 |
| dWJS | 82.7% | 0.535 | 0.319 | 69.4% | 0.537 | 0.304 | 66.1% | 0.522 | 0.294 | 85.6% | 0.545 | 0.317 |
| DiffAb | 85.5% | 0.541 | 0.317 | 86.7% | 0.548 | 0.318 | 85.6% | 0.528 | 0.317 | 84.4% | 0.540 | 0.316 |
| AbDPO | 88.3% | 0.540 | 0.318 | 91.1% | 0.545 | 0.318 | 87.8% | 0.525 | 0.319 | 90.0% | 0.540 | 0.315 |
| GearBind | 87.7% | 0.543 | 0.315 | 87.1% | 0.544 | 0.317 | 86.7% | 0.527 | 0.317 | 88.9% | 0.541 | 0.314 |
| *AffinityFlow* | **88.9%** | **0.526** | 0.320 | **93.3%** | **0.528** | 0.321 | **89.7%** | **0.514** | 0.322 | **91.2%** | **0.528** | 0.323 |

design setting, *IMP* drops from 64.0% to 22.7% for ESM, from 88.5% to 49.4% for AbLang, from 81.6% to 46.7% for nanoBERT, from 66.1% to 23.9% for dWJS, from 85.6% to 49.4% for DiffAb, from 87.8% to 50.6% for AbDPO, from 86.7% to 50.6% for GearBind, and from 89.7% to 68.3% for our method. In this context, our method demonstrates a clear advantage over the comparison methods.

**Additional Comparisons.** We further conduct experiments with gg-dWJS (Ikram et al., 2024b), employing the trained affinity predictor to guide sampling. Specifically, for the CDR-H3 region, gg-dWJS achieves IMP, Sim, and Nat scores of 67.2, 0.520, and 0.291, respectively, which are still worse than our method. We observe that the IMP score does not significantly improve compared to the original dWJS (which is 66.1). This suggests that the affinity predictor used in the post-selection step of the original dWJS already contributes effectively to guided generation.

Additionally, we implement an MCMC variant: At each step, we use ESM to identify the top 20 most probable mutations, randomly choose one mutation as a proposal, and use the affinity predictor to compute its acceptance probability. We repeat this procedure for a total of 9 steps for consistency. Evaluating the resulting sequences, we obtain scores for IMP, Sim, and Nat of 65.6, 0.562, and 0.360, respectively. While the IMP score slightly improves (from the original 64.0 to 65.6), it remains inferior to our proposed method. The drop in Sim likely stems from strong antigen-specific guidance every step, while Nat improves due to MCMC's conservative acceptance.

### 4.5. Ablation Studies

We use *AffinityFlow* as the baseline to evaluate the effect of removing specific modules, with results shown in Table 2. The ablation studies are conducted on CDR-H3, considering 10 antigens for efficiency. Our focus is primarily on the *IMP* metric, so the discussion centers around this metric.

Table 2: Ablation Study of AffinityFlow on CDR-H3.

| Methods | IMP↑ | Sim↓ | Nat↑ |
|---|---|---|---|
| *one-iteration* | 73.3% | **0.512** | 0.319 |
| *w/o PC* | 83.3% | 0.521 | 0.316 |
| *w/o AlphaFlow* | 63.3% | 0.528 | 0.314 |
| *w/o energy* | 66.7% | 0.531 | 0.322 |
| *w/o selection* | 76.7% | 0.523 | 0.326 |
| Ours | **93.3%** | 0.514 | **0.330** |

**Alternating Optimization** This framework alternates between updating the structure with the sequence fixed, and mutating the sequence with the structure fixed. In this study, we perform a single iteration, applying multiple mutations simultaneously, referred to as *one-iteration*. We also evaluate the impact of the predictor-corrector technique by excluding the Amber relaxation step, denoted *w/o PC*. As shown in Table 2, both ablations reduce performance, demonstrating the predictor-corrector and Amber relaxation effectiveness.

We also evaluate the effect of directly removing the AlphaFlow framework. In this variant, we perform gradient optimization on the existing protein structure instead of using predictor guidance. This step is followed by Amber relaxation, after which we use ProteinMPNN to identify potential mutations. This variant is denoted by *w/o AlphaFlow* in Table 2, which shows that leaving out AlphaFlow leads to the greatest performance drop compared to the other two variants. We attribute this to AlphaFlow's ability to capture the natural fluctuations of proteins, resulting in more realistic structures than those generated through direct gradient ascent alone, and accessing binding conformations that may be different from either a structure determined experimentally or predicted by a model like AlphaFold. Less realistic structures in turn yield less natural mutated sequences, as reflected by the *Nat* score decreasing from 0.330 to 0.314.

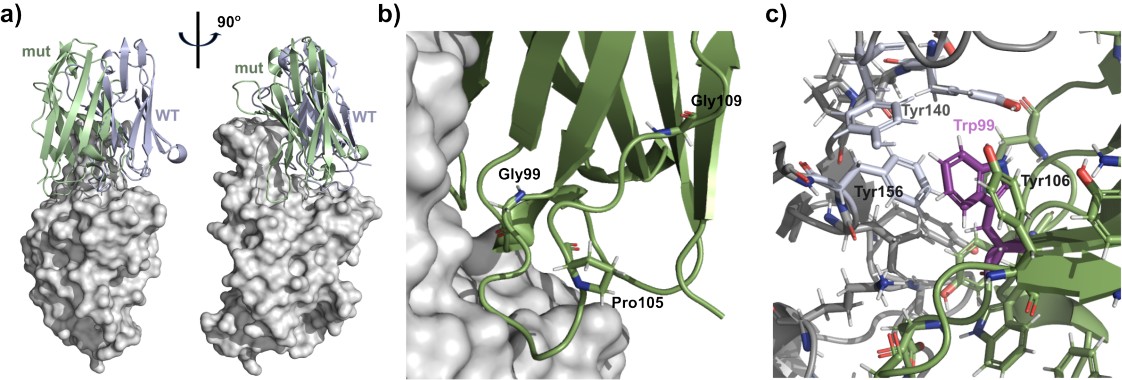

Figure 3: Visualizations of model-generated antibody structures bound to the SARS-CoV-2 RBD **(a)** Relative to a fixed antigen, most model-generated antibodies (green) are predicted to bind with a noticeable rotation in binding pose compared to the WT conformation (blue). **(b)** Our model suggests several mutations frequently, in particular Ala105Pro may stabilize the CDR loop. **(c)** The buried Lys99Trp mutation interacts with multiple other aromatic residues across the interface.

**Co-teaching** We evaluate the co-teaching module with two variants: (1) *w/o energy*: using the trained predictor on limited labeled data only. (2) *w/o selection*: training on pairwise discrete data without sample selection. As shown in Table 2, both variants reduce the *IMP* metric, highlighting the effectiveness of the module. Notably, *w/o energy* performs worse than *w/o selection*, demonstrating the value of biophysical energy data. We also observe that a better-trained predictor improves specificity: our method achieves the best *Sim*, while *w/o selection* ranks second. This likely results from the predictor's role in estimating antigen-specific binding energy, leading to greater specificity.

Additionally, we report the Spearman's rank correlation coefficient $R$ (Spearmanr) on the test set. We isolate 10 antigens from the total dataset, with each antigen paired with 77 sdAbs. We calculate Spearman's $R$ for each antigen and present the average across the 10 antigens. The models without energy data achieve $R$ values of 0.0956 for the sequence-based predictor and $-0.0043$ for the structure-based predictor, respectively, reflecting the limitations of the 120 labeled entries. By utilizing biophysical energy data for direct fine-tuning, the sequence-based predictor reaches a coefficient of 0.40, while the structure-based predictor achieves 0.50. While not state-of-the-art for antibody binding energy prediction in general, these values demonstrate the effectiveness of our approach when limited data is available. Sample selection further improves performance, with the sequence-based predictor achieving a coefficient of 0.51 and the structure-based predictor reaching 0.52. These results highlight the benefits of using biophysical energies and sample selection to enhance prediction accuracy.

### 4.6. Case Study

To further understand how *AffinityFlow* generates mutations to improve binding, we analyze the structures of our proposed mutants and the wild-type of a single-domain antibody (sdAb) known to bind the SARS-CoV-2 receptor-binding domain (RBD) (Yao et al., 2021). We generate 30 mutated structures, with half containing mutations only in the CDR3 loop and half having mutations across all CDRs. We use Rosetta to calculate binding energies ($\Delta\Delta G$) and other interface metrics relative to the wild-type structure (PDB ID 7D30).

All computed structures show $\Delta\Delta G < 0$, suggesting that the designed antibodies bind the antigen more tightly than the native sequence. However, we do not observe any correlation between $\Delta\Delta G$ and which CDRs are allowed to mutate. We measure other interface metrics (dSASA, shape complementarity) for all 30 structures and compare these values with those computed for native antibody-antigen interfaces in the PDB (Adolf-Bryfogle et al., 2018). Both metrics indicate interface quality: (1) dSASA (change in solvent-accessible surface area) reflects how well hydrophobic residues are buried and how closely the antibody and antigen interact, and (2) shape complementarity measures how well the two proteins fit together. The results align well with natural structures, demonstrating that our model preserves the correct shape profile of the binding surface. Interestingly, despite conserving the binding interface shape, most mutants (21/30) dock with a rotated binding pose of approximately 67 degrees (Figure 3a). This rotation shifts interactions away from CDR1 and toward stronger interactions in CDR2 and CDR3.

Certain mutations occur frequently across all model-

proposed antibodies, indicating that the model focuses attention on these residues. Notably, Lys99Gly, Ala105Pro, and Asp109Gly appear often, regardless of whether mutations are restricted to the CDR3 loop or allowed across all positions (Figure 3b). We believe that the Ala105Pro mutation stabilizes the CDR3 loop into an optimal conformation for this antigen. We used scikit-learn's RandomForestRegressor with 100 decision trees, training on mutation types (input) and Rosetta-predicted $\Delta\Delta G$ (output). Model was validated using R² on a $20\%$ held-out test set. We find that the rarer Ala105Leu mutation contributes most to improving $\Delta\Delta G$, likely by increasing hydrophobicity at the interface and promoting assembly.

Most intriguingly, one model-generated sequence includes a Lys99Trp mutation, an unusual amino acid to insert within an interface. Visual examination reveals that this tryptophan residue is inserted such that it creates $\pi$-$\pi$ interactions with two aromatic residues across the interface as well as providing stabilizing interactions with a tyrosine on the antibody itself (Figure 3c). This mutation is especially interesting, as we detail below. Our case study uses the MR17 nanobody (PDB: 7D30) from (Yao et al., 2021), which has a reported KD of 83.7 nM. (Li et al., 2020) later introduced a mutant, MR17m, with a Lys99Tyr substitution that improved IC50, indicating higher potency (i.e., requiring less antibody to achieve the same effect). They also suggested Lys99Trp could be even more effective—an uncommon mutation that AffinityFlow independently identified.

Our structural analysis of mutant and wild-type antibody structures reveals several key insights into the nature of mutations governing antibody-antigen binding. These results validate our computational approach and also highlight its potential to guide rational design of improved antibodies against SARS-CoV-2 and other pathogens, opening new avenues for therapeutic development.

## 5. Conclusion

We present *AffinityFlow* for optimizing antibody sequences, introducing several key innovations. First, we develop an AlphaFlow-based *alternating optimization* framework that leverages predictor guidance to steer structure generation toward high binding affinity, followed by targeted sequence mutations. Second, we propose a *co-teaching* module that integrates insights from noisy biophysical energies to refine both structure- and sequence-based predictors. Our method achieves state-of-the-art performance in affinity maturation experiments across functionality, specificity, and rationality metrics, demonstrating the effectiveness of *AffinityFlow* in advancing antibody sequence design.

## Impact Statement

Antibody affinity maturation aims to enhance the binding affinity of antibodies to their target antigens, which has broad implications in therapeutic development. This research has the potential to significantly improve the efficacy of antibody-based treatments for various diseases, including cancer, autoimmune disorders, and infectious diseases. For instance, optimizing antibodies against emerging pathogens could play a crucial role in mitigating future pandemics and saving millions of lives. While this work offers substantial societal benefits, we acknowledge the potential for dual-use concerns. Advances in antibody affinity maturation could, in principle, be misused to develop harmful applications, such as targeting specific biomolecules for malicious purposes. As researchers, we are committed to raising awareness of these risks and promoting ethical use of these methods. We firmly believe that the potential benefits of this research far outweigh the risks, given its promise to address critical global health challenges. Additionally, we emphasize the importance of community oversight and regulatory frameworks to mitigate misuse. In this study, we have focused solely on advancing machine learning methodologies for antibody optimization, and we do not foresee any immediate ethical concerns associated with this work.

## Acknowledgement

We thank Everado Villasenor Rodriguez for infrastructure support. We are grateful to Maxwell Bates and Jonah Noh from Amazon for their assistance with structure prediction. We also thank Steven Atkinson, Hao Song, and Di Sun from Amazon for their valuable discussions and feedback.

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

## A. Related Work

**Generative Protein Modeling**  Generative protein modeling primarily includes sequence-based language models and structure-based score generative models. Language models are trained on protein sequence datasets using masked prediction (Rives et al., 2019) or auto-regressive prediction (Ferruz et al., 2022). These models are often fine-tuned for specific domains like antibodies, with examples including AbLang (Tobias H. Olsen & Deane, 2022), AntiBERTa (Leem et al., 2022), IgLM (Shuai et al., 2021), and nanoBERT (Hadsund et al., 2024). Language models have also been explored for modeling tokenized protein structures (Hayes et al., 2024; Su et al., 2023).

Score-based models, such as diffusion-based and flow matching models, mainly focus on generating protein structures. **(1)** Diffusion-based models like RFdiffusionAA (Krishna et al., 2024) and AlphaFold3 (Abramson et al., 2024) generate structures through coordinate denoising. RFdiffusionAA has been applied to antibody design (Bennett et al., 2024), but its code is not open-sourced. Chroma (Ingraham et al., 2023) introduces property-specific guidance into diffusion models but does not research antibody design. Similarly, (Kulytė et al., 2024) incorporates force-field guidance but struggles to capture realistic structures due to the simplicity of the diffusion model. **(2)** Flow matching models have shown greater effectiveness and efficiency compared to diffusion models. Recent studies like AlphaFlow (Jing et al., 2024) and FoldFlow-2 (Huguet et al., 2024) explore sequence-conditioned flow matching for protein structure generation. In this work, we utilize the AlphaFlow framework for antibody sequence design due to its demonstrated effectiveness. It is worth noting that score-based generative models have also been applied to model discrete biological sequences (Campbell et al., 2024; Frey et al., 2023; Li et al., 2024; Ikram et al., 2024a).

**Co-teaching**  Co-teaching (Han et al., 2018) is a robust technique for addressing label noise by utilizing two collaborative models. Each model identifies small-loss samples from a noisy mini-batch to train the other. Co-teaching is conceptually related to decoupling (Malach & Shalev-Shwartz, 2017) and co-training (Blum & Mitchell, 1998), as all these approaches involve collaborative learning between two models. In this study, we adapt co-teaching to work with biophysical binding energy data rather than a noisy dataset. Specifically, the sequence-based predictor identifies clean samples for training the structure-based predictor, and vice versa.

## B. Predictor Guidance in Flow Matching

According to Lemma1 in (Zheng et al., 2023),

$$\tilde{v}(\boldsymbol{x}_t, t, \Delta G; \boldsymbol{\theta}) = a_t \boldsymbol{x}_t + b_t \nabla_{\boldsymbol{x}_t} \log p_{\boldsymbol{\beta}}(\boldsymbol{x}_t, t \mid \Delta G) \tag{10}$$

Based on this, we can derive:

$$\begin{aligned}
\tilde{v}(\boldsymbol{x}_t, t, \Delta G; \boldsymbol{\theta}) &= a_t \boldsymbol{x}_t + b_t \nabla_{\boldsymbol{x}_t} \log p_{\boldsymbol{\beta}}(\boldsymbol{x}_t, t) + b_t \nabla_{\boldsymbol{x}_t} \log p_{\boldsymbol{\beta}}(\Delta G \mid \boldsymbol{x}_t, t) \\
&= \tilde{v}(\boldsymbol{x}_t, t; \boldsymbol{\theta}) + b_t \nabla_{\boldsymbol{x}_t} \log p_{\boldsymbol{\beta}}(\Delta G \mid \boldsymbol{x}_t, t)
\end{aligned} \tag{11}$$

In our case, $b_t = \frac{1-t}{t}$, and this leads to:

$$\tilde{v}(\boldsymbol{x}_t, t, \Delta G; \boldsymbol{\theta}) = \hat{v}(\boldsymbol{x}_t, t; \boldsymbol{\theta}) + \frac{1-t}{t} \nabla_{\boldsymbol{x}_t} \log p_{\boldsymbol{\beta}}(\Delta G \mid \boldsymbol{x}_t, t). \tag{12}$$

## C. Computation Approximation

The guided vector field is defined by:

$$\tilde{v}(\boldsymbol{x}_t, t, \Delta G; \boldsymbol{\theta}) = \hat{v}(\boldsymbol{x}_t, t; \boldsymbol{\theta}) - \gamma \frac{1-t}{t} \nabla_{\boldsymbol{x}_t} \hat{f}_{\boldsymbol{\beta}}(\hat{\boldsymbol{x}}_1(\boldsymbol{x}_t)). \tag{13}$$

We compute $\nabla_{\boldsymbol{x}_t} \hat{f}_{\boldsymbol{\beta}}(\hat{\boldsymbol{x}}_1(\boldsymbol{x}_t))$ as:

$$\nabla_{\boldsymbol{x}_t} \hat{f}_{\boldsymbol{\beta}}(\hat{\boldsymbol{x}}_1(\boldsymbol{x}_t)) = \frac{\partial \hat{f}_{\boldsymbol{\beta}}(\hat{\boldsymbol{x}}_1(\boldsymbol{x}_t))}{\partial \hat{\boldsymbol{x}}_1} \frac{\partial \hat{\boldsymbol{x}}_1(\boldsymbol{x}_t)}{\partial \boldsymbol{x}_t} \tag{14}$$

As $t$ approaches 1, $\hat{\boldsymbol{x}}_1$ closely approximates $\boldsymbol{x}_t$, allowing for the simplification:

$$\frac{\partial \hat{\boldsymbol{x}}_1(\boldsymbol{x}_t)}{\partial \boldsymbol{x}_t} \approx \boldsymbol{I}, \tag{15}$$

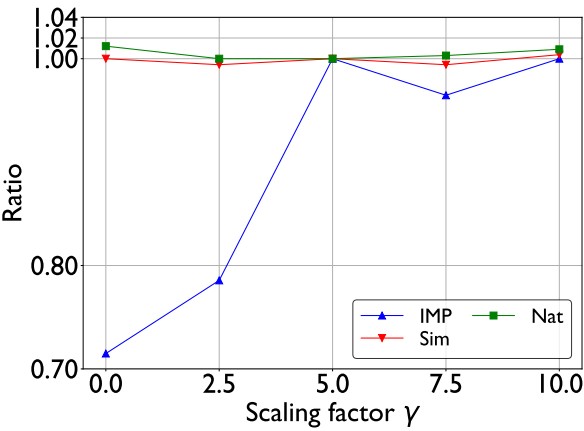

Figure 4: The antibody metrics versus scaling factor $\gamma$, normalized to their values at $\gamma$ **to** those with $\gamma = 5.0$.

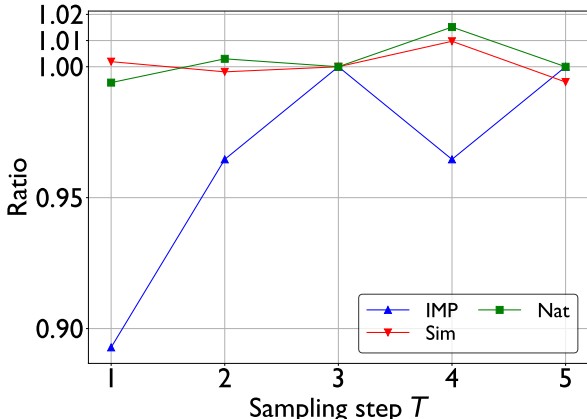

Figure 5: The three antibody metrics versus scaling factor $T$, normalized to their values at $T = 3$.

where $\boldsymbol{I}$ represents the identity matrix. Consequently, we approximate the gradient as:

$$\nabla_{\boldsymbol{x}_t} \hat{f}_{\boldsymbol{\beta}}(\hat{\boldsymbol{x}}_1(\boldsymbol{x}_t)) \approx \frac{\partial \hat{f}_{\boldsymbol{\beta}}(\hat{\boldsymbol{x}}_1)}{\partial \hat{\boldsymbol{x}}_1} \tag{16}$$

## D. Computational Efficiency

All experiments are conducted on a g5.24xlarge server equipped with GPUs with 23GB of memory. One iteration of our alternating optimization framework takes approximately 10 minutes for a protein of length 500. Language model-based methods are more computationally efficient, with processing times of 18.3 seconds for ESM, 13.0 seconds for Ablang, and 11.4 seconds for nanoBert per sample. However, our method consistently produces significantly better designs than these methods, as discussed. In applications such as antibody design, the most time-consuming and costly stage is often the evaluation of properties in wet-lab experiments. Thus, the differences in computation time between methods for generating high-performance designs are less significant in practical production settings, where optimization performance is prioritized over computational speed. This is consistent with the discussions in A.7.5 Computational Cost (Chen et al., 2023).

## E. Hyperparameter Analysis

This section examines the sensitivity of our method to various hyperparameters—namely, the scaling factor ($\gamma$) and the number of sampling steps ($T$) on CDR-H3 with 10 antigens. The reported metrics are normalized by dividing by the default hyperparameter result to facilitate comparative analysis.

**Scaling Factor ($\gamma$):** The effect of varying $\gamma$ is investigated with values 0.0, 2.5, 5.0, 7.5, and 10, and $\gamma = 5.0$ as the standard setting. As indicated in Figure 4, the *Sim* and *Nat* metrics are stable across the range of $\gamma$. However, below $\gamma \sim 5.0$ the *IMP* metric drops substantially, presumably because there is insufficient exploration of alternate backbone conformations when $\gamma$ is small.

**Number of Sampling Steps ($T$):** We analyze the impact of the number of sampling steps $T$ on the effectiveness of our method. The normalized metric is plotted as a function of $T$ in Figure 5. Again the *Sim* and *Nat* metrics are relatively unaffected by the choice of $T$, but *IMP* requires 3 sampling steps for maximum benefit. Again this suggests that more exploration of the conformational space improves the final design.

