# OpenReview forum: "AffinityFlow: Guided Flows for Antibody Affinity Maturation"
_ICML.cc/2025/Conference — ICML 2025 poster_

### Official Review · Reviewer_QVPq · 2025-03-09

**Overall Recommendation:** 4

**Summary:**

This manuscript proposes an alternating optimization framework for designing antibodies. In the first stage of the cycle, for a given fixed sequence, structures are generated with high binding affinity using a (structure-based) predictor guidance of AlphaFlow. In the second, the structures are inverse-folded to create mutated sequences and selected by a sequence-based affinity predictor. The cycles may be repeated to accumulate mutations. A key feature of the framework is co-teaching of the structure-based and sequence-based affinity predictors, based on selecting a subset of generated instances based on prediction consensus and updating the predictors based on just the selected subset. Experiments demonstrate superior affinity improvement and antigen specificity and competitive naturalness relative to methods based on protein language models trained on a large corpus, a sequence-based generative model, and structure-based generative models.

## update after rebuttal
The authors have incorporated my comments, importantly ones concerning accessibility to the general non-bio audience. The new experiments with gg-dWJS are convincing. I've raised my score to an accept.

**Claims And Evidence:**

The paper claims "state-of-the-art performance in affinity maturation" but only uses the in silico $\Delta \Delta G$ energies for guidance and evaluation. $\Delta \Delta G$ has often been shown to correlate poorly with affinity (as measured by the dissociation constant KD), depending on the Rosetta protocol used. Mason et al. 2021, for instance, did not observe a significant correlation, and there are efforts to improve Rosetta protocols to improve the agreement (Das et al. 2017). The key challenge in affinity maturation originates from the poor mapping between in silico proxies and experimental KD; without addressing this, I believe the paper's claim should be phrased less strongly than SOTA in "affinity maturation." While guided generation based on $\Delta \Delta G$ is still interesting and obtaining experimental KD data would be very expensive, the experiment should be considered a proof-of-concept of the proposed framework rather than a biologically significant scientific result.

```
Mason, Derek M., et al. "Optimization of therapeutic antibodies by predicting antigen specificity from antibody sequence via deep learning." Nature biomedical engineering 5.6 (2021): 600-612.

Dias, Raquel, and Bryan Kolaczkowski. "Improving the accuracy of high-throughput protein-protein affinity prediction may require better training data." BMC bioinformatics 18 (2017): 7-18.
```

**Essential References Not Discussed:**

There is a substantial body of work in predictor guidance for both sequence-based and structure-based generative models. Please see the references in Meng et al. 2024 for a comprehensive list.

```
Meng, Fanxu, et al. "A comprehensive overview of recent advances in generative models for antibodies." Computational and Structural Biotechnology Journal (2024).
```

**Experimental Designs Or Analyses:**

- While AffinityFlow seems to outperform baselines in functionality and specificity, the metric values are quite close and it is difficult to assess significance without repeated runs.
- The ablation study is comprehensive and makes it clear that all components of the framework (multiple iterations, predictor-corrector, AlphaFlow, biophysical energy data, and selection) are helpful.
- The one sequence-based generative model baseline (dWJS) is unconditional and does not use any predictor guidance. For a "fair" comparison, dWJS should only be trained on the $\Delta \Delta G < 0$ instances.
- Readers not familiar with antibody design will have trouble understanding the paper, as it does not introduce domain-specific concepts. Examples: change in binding free energy (what it being negative means), CDR and its regions, how many amino acids there are (not mentioned).
- Because the metrics are all in silico, Section 4.6 (Case Study) is especially important for making the case that the generated designs are biologically meaningful. But it is currently very difficult to read without domain expertise. Details of the random forest regression are not described, even in the Appendix. Please describe thoroughly the connections to prior experimental studies such as Li et al. 2020.
- What value does the structure-based predictor $\hat f_\beta$ predict?
- There is no discussion of the form of the sequence-based and structure-based predictors.

**Methods And Evaluation Criteria:**

SAbDab-nano is a widely used dataset of antibody structures for sdAb. Functionality, specificity, and rationality (naturalness) are reasonable metrics capturing complementary aspects of generated designs. For functionality, it would have been instructive to evaluate on test $\delta \delta G $ values based on a few other Rosetta protocols.

**Other Comments Or Suggestions:**

N/A

**Other Strengths And Weaknesses:**

- While the paper is creative and grounded in its combination of existing ideas, it is currently very difficult for a non-antibody expert to read. Please see "Experimental Designs Or Analyses" regarding the strengths and weaknesses of the experiments, questions about experimental details, and suggestions for improving the exposition for the lay audience.
- Please see "Claims And Evidence" for a discussion of the interpretation of Rosetta energies.

**Questions For Authors:**

Please see "Experimental Designs Or Analyses" regarding the strengths and weaknesses of the experiments, questions about experimental details, and suggestions for improving the exposition for the lay audience.

**Relation To Broader Scientific Literature:**

This work borrows from the latest developments in structural ensemble generation methods such as AlphaFlow, inverse folding (derivatives of ProteinMPNN), and guided generation (predictor guidance). It combines techniques from each of these fields under a single pipeline, to generate high-affinity antibodies in a manner informed by both sequence and structure. Simple tricks like consensus-based sample selection and correction using Amber relaxation are important contributions that improve performance.

**Theoretical Claims:**

The paper does not make any theoretical claim.

---

> ### Author Rebuttal · Authors · 2025-03-31
>
> ## General Reply
>
> Thank you for your insightful comments—they’ve greatly improved our manuscript. We’ve addressed each point and will update the manuscript accordingly.
>
> ## Claims and Evidence
>
> > poor mapping
>
> Thank you for your feedback. We agree that the gap between in silico proxies and experimental KD is a key challenge. Accordingly, we will revise the abstract to state: "Our method, AffinityFlow, achieves state-of-the-art performance in proof-of-concept affinity maturation experiments."
>
> ## Experimental Designs Or Analyses
>
> > difficult to assess significance
>
> We acknowledge that AlphaFlow's inference involves computationally expensive protein conformation modeling, which limited our ability to perform repeated runs. Nonetheless, the reported performance differences, though modest, consistently favor AffinityFlow in functionality and specificity.
>
> > dWJS should be trained on ddG<0.
>
> As noted in Section 4.2, we use a sequence-based predictor to select top candidates for dWJS, effectively conditioning dWJS. Moreover, dWJS is trained solely on antibody sequences without antigen context, per the original paper, so filtering by ddG < 0 is not applicable. We also compare the guided version (gg-dWJS) in QKfP's rebuttal, which still underperforms our method.
>
> > trouble understanding
>
> We have described the change in binding free energy as the difference between the free energies of the bound and unbound states in Section 2.4 (Lines 127–130). A negative value indicates that the overall free energy of the system decreases upon binding, meaning that the antibody–antigen interaction is energetically favored. We will add these in Section 2.4 (Line 127).
>
> An antibody consists of two heavy chains and two light chains with a similar overall structure. Its specificity is determined by six variable regions known as Complementarity Determining Regions (CDRs), denoted as H1, H2, H3, L1, L2, and L3. Typically, heavy chain CDRs range from $8$ to $16$ amino acids, while light chain CDRs range from $3$ to $10$ amino acids. We will include these in Section 2.1 (Line 80).
>
> > difficult to read
>
> (1) Random forest:  We used scikit-learn’s RandomForestRegressor with 100 decision trees, training on mutation types (input) and Rosetta-predicted ΔΔG (output). Model was validated using R² on a 20\% held-out test set. Feature importances revealed Ala105Leu as the most influential mutation.
>
> (2) Connections to prior studies: Our case study uses the MR17 nanobody (PDB: 7D30) from Yao et al. (2021), which has a reported KD of 83.7 nM. Li et al. (2021) later introduced a mutant, MR17m, with a Lys99Tyr substitution that improved IC50, indicating higher potency (i.e., requiring less antibody to achieve the same effect). They also suggested Lys99Trp could be even more effective—an uncommon mutation that AffinityFlow independently identified.
>
> We will incorporate these discussions into Section 4.6.
>
> > predictor predict
>
> The structure-based predictor outputs the negative value of the binding affinity (Kd). We will add this clarification in Section 2.4 (Line 131).
>
> > form of predictors
>
> We describe the architectures of both the sequence-based and structure-based predictors in Section 2.4, with detailed hyperparameters provided in Section 4.3. To further clarify, we will include more details regarding the MLP and GVP head in Section 4.3.
>
> ## Essential References Not Discussed
>
> > see the references in Meng et al. 2024.
>
> Thank you for highlighting this survey. We have reviewed Section 2.2.3 ("Hybrid generative models") of Meng et al. (2024) and confirm that key methods such as DiffAb and Chroma have been discussed in our paper. We note, however, that most methods listed in Meng et al. (2024) do not specifically address affinity maturation. To clarify, we will add to Appendix A: "ABGNN pre-trains a novel antibody language model and introduces a one-shot approach for generating both sequence and structure of CDRs. AbDiffuser leverages domain knowledge and physics-based constraints to enhance diffusion modeling. AlphaPanda integrates transformer, 3DCNN, and diffusion models for joint sequence-structure co-design. A more detailed overview can be found in Meng et al. (2024). Notably, these methods primarily focus on general antibody design rather than specifically targeting affinity maturation."
>
> ## Other Strengths And Weaknesses
>
> > very difficult to read
>
> In addition to the points in “Experimental Designs or Analyses”, we will add following clarifications: (1) dSASA (change in solvent-accessible surface area) reflects how well hydrophobic residues are buried and how closely the antibody and antigen interact; (2) Shape complementarity measures how well the two proteins fit together. Both metrics indicate interface quality.
>
> Adolf-Bryfogle et al. (2018) assessed these values for all naturally occurring antibody-antigen interfaces in PDB. The metrics we calculate using Rosetta for our designs, fit well within the distribution observed in that paper.

---

> > ### Comment · Reviewer_QVPq · 2025-04-05
> >
> > Thank you for incorporating my feedback, a bulk of them concerning accessibility to the general non-bio audience. The new experiments with gg-dWJS are convincing. I'll raise my score to an accept.

---

> > > ### Author Response · Authors · 2025-04-05
> > >
> > > Thank you for your thoughtful feedback and support! We're glad the revisions improved accessibility for a broader audience and that the gg-dWJS experiments addressed your concerns. We appreciate your decision to raise the score.

---

### Official Review · Reviewer_PaUw · 2025-03-09

**Overall Recommendation:** 3

**Summary:**

The authors propose a pipeline to optimize sequences with structural guidance. AffinityFlow builds on AlphaFlow, a sequence-conditioned generative model. They present a two-stage optimization process: first, structure generation using a fixed sequence to guide the structure toward high binding affinity, followed by inverse folding and updating the initial sequence. The authors employ a co-teaching module to update the sequence after the inverse folding step.

**Claims And Evidence:**

The claims are supported by experimental results.

**Essential References Not Discussed:**

I believe BindCraft is relevant to this work, as the authors optimize a sequence based on a structure network in an iterative process.

**Experimental Designs Or Analyses:**

The experiments seem reasonable, as the authors consider modifications on three CDR regions of an antibody and compare against multiple existing methods.

I think it would be interesting to compare against ESM combined with your sequence predictor. I believe it is possible to perform an MCMC where you use your predictor to accept or reject proposals. This could help understand which parts of the pipeline are the most important (similar to your ablation study in Table 2).

**Methods And Evaluation Criteria:**

The methods utilize established tools from the literature and combine them with a predictor. The clarity of the methods could be improved, especially regarding the training of the predictor. The evaluation appears sound, although I would suggest adding comparisons of inference time and comparing against methods that also utilize iterative optimization or refinement (if this is not already the case).

**Other Comments Or Suggestions:**

I don't have any other comments or suggestions.

**Other Strengths And Weaknesses:**

The method seems to improve on existing methods for specific tasks.

However, I think the method may require more time during inference. I would encourage the authors to report inference time and peak memory usage for each method (at least for their method).

**Questions For Authors:**

- I might have missed it, but how do you train $f_\beta$ in Eq. 7? Is it a pre-trained energy function? Writing the loss function used to train the structure-based predictor could improve clarity.

- I am slightly confused with the notation $f_\alpha$ and $f_\beta$. One is for sequences and the other for structures, correct?

- I am not familiar with all the methods used for comparison. Is there a method that also includes a refinement/optimization loop? It would be great to include one if that is not already the case, or directly compare with [1] BindCraft if possible.

[1] Pacesa, M., Nickel, L., Schellhaas, C., Schmidt, J., Pyatova, E., Kissling, L., ... & Correia, B. E. (2024). BindCraft: one-shot design of functional protein binders. bioRxiv, 2024-09.

**Relation To Broader Scientific Literature:**

The key contributions of the paper relate to other works on designing pipelines for generating new proteins while steering generation toward certain properties, such as binding affinities.

**Theoretical Claims:**

There are no theoretical claims made in the paper.

---

> ### Author Rebuttal · Authors · 2025-03-31
>
> ## General Reply
>
> Thank you for your constructive feedback, which has improved the clarity and rigor of our paper. We have addressed all points and will revise the manuscript accordingly.
>
> ## Methods And Evaluation Criteria:
> > The clarity of the methods (predictor training).
>
> To further clarify our method, we highlight the following points:
>
> (1) Predictor Architecture: We described the predictor architecture in Section 4.3, where $\alpha$ and $\beta$ represent the parameters of the sequence-based and structure-based predictors, respectively. We will explicitly reiterate these parameter definitions in Section 4.3 to improve readability.
>
> (2) Predictor Training: The training of our predictors involves two phases: (a) Supervised Training: Initially, we train both predictors using labeled antibody-antigen sequence/structure data and their affinity labels. Specifically, we optimize the parameters by minimizing the mean squared error (MSE) loss: $(f_{\alpha}(a) - \Delta G)^2$ for the sequence-based predictor and $(f_{\beta}(x) - \Delta G)^2$ for the structure-based predictor. Here $a$ and $x$ denote the protein sequence and structure respectively and $\Delta G$ corresponds to the negative of affinity. We will explicitly present these in Section 2.4. (b) Co-teaching Fine-tuning: we further refine both predictors using Rosetta-generated labeled data, optimizing the loss in Eq. (9), to further enhances performance.
>
>
> > comparisons of inference time and comparing against methods that also utilize iterative optimization or refinement (if this is not already the case).
>
> We have compared the inference time of our method with language model-based approaches in Appendix D. For methods that do not rely on language models, their iterative optimization process takes less than $10$ seconds per sample. Although these alternative methods are more efficient, in the context of antibody design, achieving high-affinity designs is prioritized over computational speed.
>
> ## Experimental Designs Or Analyses
>
> > I think it would be interesting to compare against ESM combined with your sequence predictor.
>
> Thank you for the insightful suggestion. Our current ESM baseline already incorporates the sequence predictor to select the top three sequences per antigen (Section 4.2).
>
> To further explore your idea, we implemented an MCMC variant: At each step, we use ESM to identify the top 20 most probable mutations, randomly choose one mutation as a proposal, and use the affinity predictor to compute its acceptance probability. We repeat this procedure for a total of 9 steps for consistency.
>
> Evaluating the resulting sequences, we obtain scores for IMP, Sim, and Nat of 65.6, 0.562, and 0.360, respectively. While the IMP score slightly improves (from the original 64.0 to 65.6), it remains inferior to our proposed method. The drop in Sim likely stems from strong antigen-specific guidance every step, while Nat improves due to MCMC’s conservative acceptance. We will incorporate these discussions into Section 4.2.
>
> ## Essential References Not Discussed
>
> > BindCraft relevant
>
> Thank you for highlighting this work. We will add the following to Appendix A:
>
> "BindCraft adopts AF2-multimer to generate a binder backbone and sequence given a known target protein structure, subsequently optimizing the non-interface regions using ProteinMPNN. However, BindCraft is not directly comparable to our method, as it specifically targets binder design with known target structures, whereas our setting focuses on affinity maturation through mutation of existing antibody sequences without direct access to target structures. Moreover, the primary goal of BindCraft is generating new binders, rather than improving binding affinity of existing antibodies."
>
> ## Other Strengths And Weaknesses
>
> > report inference time
>
> Our method does require more inference time due to AlphaFlow’s realistic structure modeling, as discussed in Appendix D. While less efficient than alternatives, it consistently yields better designs. In practice, where wet-lab evaluation dominates cost and time, optimization quality is prioritized over computational speed.
>
>
> ## Questions For Authors:
> > how train $f_{\beta}$ in Eq. 7
>
> We first train $f_{\beta}$ using the MSE loss and then fine-tune it with the loss in Eq. (9). It is not a pre-trained energy function; rather, it is a property predictor built upon the ESM2-GVP backbone. We have described this training procedure and loss in the rebuttal of "Methods and Evaluation Criteria".
>
>
>
> > confused with notation
>
> Yes, $f_{\alpha}(a)$ refers to the sequence-based predictor, and $f_{\beta}(x)$ refers to the structure-based predictor.
>
> > not familiar with comparison methods
>
> All non-language-model-based methods used for comparison—including dWJS, DiffAb, AbDPO, and GearBind—employ iterative optimization loops for refinement. Our results demonstrate that AffinityFlow consistently outperforms these approaches.

---

### Official Review · Reviewer_QKfP · 2025-03-10

**Overall Recommendation:** 4

**Summary:**

The work combines classifier (gradient) guidance with Alphaflow for flow matching based antibody structure optimization to enhance binding affinity and performs inverse folding with ProteinMPNN to retrieve antibody sequences for synthesis. It also proposes a noise reduction framework (co-teaching) for labeled data whilst training affinity predictors. The authors evaluate their method's performance on SAbDab dataset which shows improvements over the baselines.

**Claims And Evidence:**

Yes

**Essential References Not Discussed:**

N/A

**Experimental Designs Or Analyses:**

See Methods And Evaluation Criteria.

**Methods And Evaluation Criteria:**

Given the nature of the work which is optimization, I expect more baselines that uses optimization techniques. Methods such as ESM and AbLang are not particularly suitable for optimizing for Antigen affinity. The author's also use dWJS as a baseline. Given their method uses classifier guidance, why not use the classified guidance version of the dWJS, i.e., gg-dWJS? Given the limited nature of experiments, I think such expectation is justified.

**Other Comments Or Suggestions:**

N/A

**Other Strengths And Weaknesses:**

Strengths

- The writing in general is concise and easy to understand
- The proposed method works as claimed shown through the experiments (although I remain unconvinced about the baselines uses, see evaluation and criteria)

**Questions For Authors:**

- Given the pitfalls of inverse folding (noisiness for conversion of sequence to structure) and structure optimization (unrealizability of structures), how do you guarantee their method's generated sequences are synthesizable? Did you try synthesizing any samples suggested by your method?

- Did the authors attempt any in vitro experiments?

- Why is flow matching a better option for this work?

**Relation To Broader Scientific Literature:**

The work adds classifier guidance to flow based method for antibody sequences and uses inverse folding to convert to sequence (so that they could be synthesized).

**Theoretical Claims:**

N/A

---

> ### Author Rebuttal · Authors · 2025-03-31
>
> ## General Reply
>
> Your insightful comments have greatly contributed to improving our manuscript, and we sincerely appreciate your time and effort. Each point you mentioned has been addressed, and the manuscript will be updated to reflect these improvements.
>
> ## Methods And Evaluation Criteria:
> > Given the nature of the work which is optimization, I expect more baselines that uses optimization techniques. Methods such as ESM and AbLang are not particularly suitable for optimizing for Antigen affinity. The author's also use dWJS as a baseline. Given their method uses classifier guidance, why not use the classified guidance version of the dWJS, i.e., gg-dWJS? Given the limited nature of experiments, I think such expectation is justified.
>
> As discussed in Section 4.2, we employ the same trained sequence-based affinity predictor for final sequence selection across all baselines, including ESM, AbLang, and dWJS. This strategy inherently enables affinity optimization, making these methods valid baselines for comparison.
>
> Thank you for pointing out gg-dWJS. To clarify, we have further conducted experiments with gg-dWJS, employing the trained affinity predictor to guide sampling, following the original gg-dWJS paper. Specifically, for the CDR-H3 region, gg-dWJS achieves IMP, Sim, and Nat scores of 67.2, 0.520, and 0.291, respectively, which are still worse than our method. We observe that the IMP score does not significantly improve compared to the original dWJS (which is 66.1). This suggests that the affinity predictor used in the post-selection step of the original dWJS already contributes effectively to guided generation. We will incorporate this additional discussion into Section 4.4 to clarify this point further.
>
> ## Questions For Authors:
>
> > Given the pitfalls of inverse folding (noisiness for conversion of sequence to structure) and structure optimization (unrealizability of structures), how do you guarantee their method's generated sequences are synthesizable? Did you try synthesizing any samples suggested by your method?
>
> We address the concern regarding synthesizability from three key perspectives:
>
> (a) Realistic Structure:  (1) We use AlphaFlow as our protein structure generation framework, which has demonstrated its effectiveness in producing realistic protein conformations.   (2) In addition, our Predictor-Corrector method incorporates Amber relaxation at every iteration to refine the protein coordinates, ensuring the generated structures are physically realistic.
>
> (b) Limited Mutation Per Iteration: (1) Although the inverse folding process (mapping structure to sequence) can be noisy, we mitigate this issue by restricting mutations to only 1-3 positions per stage rather than generating the entire sequence at once. This targeted approach reduces noise; (2) Furthermore, we employ a post-selection process using a trained sequence-based predictor to filter out sequences with low predicted affinity.
>
> (c) Biologically Meaningful Case Study:
> We present a detailed case study in Section 4.6 to demonstrate the biological relevance of our generated designs. Notably, our AffinityFlow model independently identified the Lys99Trp mutation—a mutation previously proposed in Li et al. (2020) as a promising improvement in antibody potency.
>
> These strategies collectively enhance the likelihood that our generated sequences are synthesizable and biologically meaningful. Furthermore, we report Nat (the inverse of perplexity) as a supporting metric. As shown in Table 1, AffinityFlow achieves the highest Nat score among all non-language model-based methods, indicating stronger sequence plausibility.
>
>
>
> > Did the authors attempt any in vitro experiments?
>
> We have not conducted in vitro experiments at this stage.
>
> > Why is flow matching a better option for this work?
>
> We adopt flow matching primarily due to the availability of a pre-trained AlphaFlow framework, which has demonstrated exceptional performance in generating realistic protein conformations. Additionally, recent studies [1, 2, 3] show that flow matching can offer superior effectiveness and efficiency compared to diffusion models.
>
> It is also important to note that our proposed alternative framework and co-teaching module are not inherently tied to the AlphaFlow framework; they can, in principle, be implemented within any sequence-conditioned generative model of structure.
>
>      - [1] Lipman Y, Chen R T Q, Ben-Hamu H, et al. Flow matching for generative modeling[J]. arXiv preprint arXiv:2210.02747, 2022.  ICLR 2023
>      - [2] Le M, Vyas A, Shi B, et al. Voicebox: Text-guided multilingual universal speech generation at scale[J]. NeurIPS 2023.
>      - [3] Adam Polyak, et al. Movie gen: A cast of media foundation models. 2024.

---

> > ### Comment · Reviewer_QKfP · 2025-04-05
> >
> > Thanks for your efforts. I have updated my scores to an accept. Please incorporate the changes to your manuscript.
> > - "Why is flow matching a better option for this work?" -> the answer to this should be added to the motivation in the intro.
> > - A note on synthesizability somewhere in the conclusion.
> > - The results on the gg-dWJS. Minor: the citation could be updated [1]
> >
> >
> >
> >
> > 1. Ikram, Zarif, Dianbo Liu, and M. Saifur Rahman. "Gradient-guided discrete walk-jump sampling for biological sequence generation." Transactions on Machine Learning Research.

---

> > > ### Author Response · Authors · 2025-04-05
> > >
> > > Thank you for your thoughtful feedback and support! We’re glad to hear you’ve updated your score to an accept. We will incorporate the requested clarifications into the introduction and conclusion, update the gg-dWJS citation, and ensure all revisions are reflected in the final manuscript.

---

### Official Review · Reviewer_wAhe · 2025-03-14

**Overall Recommendation:** 4

**Summary:**

The paper proposes the AffinityFlow model and constructs an optimization framework for generating high-affinity antibodies. First, it utilizes a structure-based affinity predictor to guide the generation of antibody structures. Subsequently, it creates sequence mutations through inverse folding. This model enables a sequence-conditioned generative model of structure, iteratively guiding the generation of high-affinity antibodies. It has been validated on the SAbDab dataset.

## Update After Rebuttal
Thank you for the authors' response, which has resolved my confusion. I have increased my score.

**Claims And Evidence:**

I think the architecture of the model is not clearly described. There is relatively little discussion about the Predictor-Corrector and Sequence Mutation.

**Essential References Not Discussed:**

I think the relevant literature has been covered.

**Experimental Designs Or Analyses:**

I think the experimental design is reasonable.

**Methods And Evaluation Criteria:**

It is reasonable to conduct the evaluation and select the indicators on the SAbDab dataset.

**Other Comments Or Suggestions:**

There are no other suggestions here.

**Other Strengths And Weaknesses:**

Pros:

The process of the alternating algorithm model generating high-affinity is both interesting and reasonable.

Cons:
1. The description of model architecture is not very clear. Is the model end-to-end? How are the ESM-2, ESM2-GVP, AlphaFlow, etc. used organized? An algorithm or a framework of the model is needed for demonstration.

2. The SAbDab dataset is small. How can we ensure that the model does not overfit?

3. The paper lacks comparative experiments on runtime.

If you can clear up my confusion, I will raise my score because the experimental results are objective.

**Questions For Authors:**

All the questions are in "Other Strengths And Weaknesses".

**Relation To Broader Scientific Literature:**

Generating antibodies with high affinity is of great importance. However, currently, it may be limited by the relatively small amount of data with known affinity labels. Therefore, research on related artificial intelligence methods is highly necessary.

**Theoretical Claims:**

I read about Guided Structure Generation, Sequence Mutation, but I'm still confused about the overall framework of the model and how the data is connected.

---

> ### Author Rebuttal · Authors · 2025-03-31
>
> ## General Reply
>
> Thank you for your valuable feedback, which has greatly improved our manuscript. We have addressed each comment and will incorporate the revisions accordingly.
>
> ## Claims And Evidence
>
> > I think the architecture of the model is not clearly described. There is relatively little discussion about the Predictor-Corrector and Sequence Mutation.
>
> We provide additional details on both Predictor-Corrector and Sequence Mutation.
>
> (1) Predictor-Corrector is composed of two components. Predictor corresponds to the protein coordinate generation process governed by the learned vector field, and Corrector refers to the Amber energy minimization used to refine the coordinates. We will add the sentence in Line 180.
>
> (2) Sequence Mutation: At each iteration, we apply single-, double-, and triple-point mutations using ProteinMPNN. For each position, we calculate the probability difference between the current and alternative amino acids, selecting the mutation with the highest difference. Double- and triple-point mutations build sequentially on prior mutations. A sequence-based predictor selects the top K (K=3) sequences at each stage for further refinement. We will add this description at Line 191 for clarity.
>
> ## Theoretical Claims:
>
> >  I read about Guided Structure Generation, Sequence Mutation, but I'm still confused about the overall framework of the model and how the data is connected.
>
> To clarify the overall framework and data flow, consider this example:
>
> Given antibody and antigen sequences, our goal is to mutate the antibody to improve binding affinity. We begin by linking the antibody and antigen sequences and inputting this linked sequence into the AlphaFlow. Without predictor guidance, AlphaFlow generates conformations consistent with the linked sequence.
>
> However, our objective is to obtain a protein conformation with higher binding affinity. To achieve this, we introduce predictor guidance during the AlphaFlow structure generation phase, a process we call Guided Structure Generation. This guidance steers the generated conformations toward those with high binding affinity. Once a high-affinity structure is obtained, we apply inverse folding to introduce targeted mutations—Sequence Mutation—and feed the updated sequence back into AlphaFlow. This iterative loop continues to optimize binding affinity. This process is described between Lines 42–73.
>
> ## Other Strengths And Weaknesses:
>
> > The description of model architecture is not very clear.
>
> Yes, the model is end-to-end. Our architecture is organized as follows:
>
> (1) ESM-2 serves as the sequence-based predictor and is applied during post-selection (as shown in Figure 1) to choose high-affinity mutants for the next iteration.
> (2) ESM2-GVP acts as the structure-based predictor and is used for predictor guidance, steering the noisy protein coordinates toward high-affinity conformations.
> (3) AlphaFlow is the core algorithm, initially trained to transform noisy protein coordinates into clean conformations based on the input sequence. In our framework, AlphaFlow is further guided by ESM2-GVP to bias the generated structures toward higher binding affinity.
>
> We will update Figure 1 to clearly label ESM2, ESM2-GVP, and AlphaFlow, and include an overall algorithm that demonstrates the complete framework in the revised paper.
>
> > The SAbDab dataset is small.
>
> We address this issue with the following strategies:
>
> (1) AlphaFlow: AlphaFlow is pretrained on large-scale protein structures. In our method, this pretrained model remains frozen, thereby preventing it from overfitting the small dataset.
>
> (2) Sequence-based (ESM-2) and Structure-based Predictors (ESM2-GVP): Both predictors leverage pretrained ESM-2 models, which have been pretrained on large-scale unlabeled data. During our training, we keep these pretrained backbones frozen and fine-tune only the prediction heads. This reduces the risk of overfitting due to the small number of learnable parameters.
>
> (3) Augmented Dataset via Rosetta Labeling and Co-teaching: To further improve generalization, we introduce a co-teaching module. Initially, the predictors are trained on the limited SAbDab dataset. Next, we augment the training data by generating an additional $4,158$ labeled samples using Rosetta. The co-teaching module then mitigates noise from the augmented data, providing a richer dataset and reducing the risk of overfitting.
>
> These measures collectively ensure that our models remain robust and generalize effectively, despite the limited size of SAbDab.
>
> > lacks comparative runtime.
>
> We address runtime in Appendix D: one iteration of our method takes around $10$ minutes for a protein of length $500$, compared to $11$–$18$ seconds for language model-based methods per sample. While these alternatives are faster, our approach yields better designs. In practical settings like antibody design, wet-lab evaluation is the major bottleneck, making optimization quality more critical than generation speed.

---

> > ### Comment · Reviewer_wAhe · 2025-04-09
> >
> > Thanks for your reply. It has resolved my confusion to some extent. I've raised the score to 4. I hope you can add a detailed description of the model architecture in the camera-ready version.

---

> > > ### Author Response · Authors · 2025-04-09
> > >
> > > Thanks for your response. We're glad to hear the clarification helped. We appreciate the updated score and will make sure to include a detailed description of the model architecture in the camera-ready version.

---

### Decision · Program_Chairs · 2025-05-01

**Decision:**

Accept (poster)

**Comment:**

The authors present AffinityFlow, a pipeline for sequence optimization guided by structural information. Building upon AlphaFlow, a sequence-conditioned generative model, the method introduces a two-stage optimization process: (1) structure generation using a fixed sequence to enhance binding affinity, followed by (2) inverse folding to update the initial sequence. A co-teaching module further refines the sequence after inverse folding.

The manuscript is well-written, concise, and clearly presented. Experimental results validate the effectiveness of the proposed method. During rebuttal, the authors have satisfactorily addressed reviewers' concerns, including: (1) Providing additional architectural details; (2) Expanding baseline comparisons; (3) Incorporating essential references for broader context

Following the authors' thorough responses, all reviewers unanimously recommend acceptance of the manuscript in its current form. I therefore vote to accept this paper.